# Language Model Embeddings Can Be Sufficient for Bayesian Optimization

## Abstract

Bayesian Optimization is ubiquitous in experimental design and black-box optimization for improving search efficiency. However, most existing approaches rely on regression models which are limited to fixed search spaces and structured, tabular input features. This paper explores the use of LLM embeddings over string inputs for in-context regression in Bayesian Optimization. Our results show that representing inputs as strings enables general-purpose regression across diverse domains, including synthetic, combinatorial, and hyperparameter optimization. Furthermore, our approach achieves optimization performance comparable to state-of-the-art Gaussian Process-based methods such as Google Vizier, and demonstrates potential for broader and more flexible applications.

## 1 Introduction

A fundamental component of all *value-based* search methods is regression, in which proposed solutions are filtered by predictions on their performance, before evaluation. By utilizing an accurate regression model, or *regressor*, along with balanced explore-exploit mechanisms such those proposed in Bayesian Optimization, large improvements to the sample complexity of search have been widely possible. However, for the field of traditional optimization, many regression methods so-far have been task-specific, due to the reliance of modeling assumptions and dimensionality constraints. Common tabular regression methods such as random forests and Gaussian Processes are particularly susceptible to this issue, as their input dimensions are dependent on the problem space.

Large language models (LLMs) offer a promising approach to regression due to their ability to represent information as strings, enabling more flexible input formats than traditional methods. Unlike tabular models, LLMs are not constrained by fixed-length input representations, making them more adaptable across different tasks. This flexibility allows LLM-based regressors to generalize beyond task-specific settings, addressing one of the key limitations of existing regression techniques. If LLMs prove effective in optimization, they could have far-reaching implications for accelerating search across a wide range of evaluation functions that map strings to objective values.

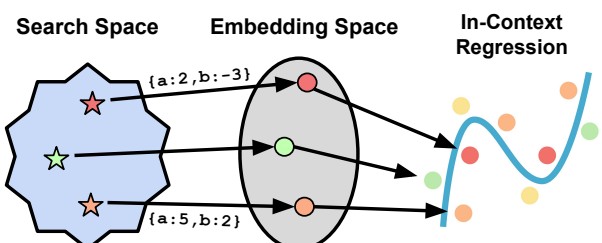

Figure 1: Using language models, we embed string representations of search space candidates as features for downstream regression.

As a starting point, in this work we focus on the well-established setting of traditional black-box optimization over tabular inputs. We explore the use of LLM-based embeddings, which map arbitrary strings into fixed-length vectors suitable for downstream tensor-based regressors that effectively balance exploration and exploitation. Specifically, our contributions are as follows:

- Our framework "embed-then-regress" describes a simple recipe for using free-form string representations in Bayesian Optimization, by using language models to embed trial inputs as features for in-context regression.

- By using a standard Transformer Neural Process (Nguyen & Grover, 2022) as the downstream regressor and pretraining it at scale over a large variety of offline evaluation data, we can achieve accurate and well-calibrated numeric predictions over unseen objective functions, competitive with specialized, state-of-the-art methods.

- After augmenting the framework with explore-exploit techniques, we achieve competitive optimization results over a variety of optimization tasks, including synthetic, combinatorial, and hyperparameter optimization, in particular matching the performance of industry-grade traditional Gaussian Process techniques such as Google Vizier.

## 2 Related Work and Motivation

Bayesian Optimization refers to a class of techniques which use regressors for solving non-differentiable optimization problems, by suggesting best candidates according to an explore-exploit tradeoff. For a given objective, the speed of optimization relies heavily on the regressor's underlying *prior*, or assumptions about the nature of the objective, such as its smoothness and landscape. Largely dominated by the use of Gaussian Process (GP) regressors, the field of Bayesian Optimization has thus seen a rise in works (Wang et al., 2024; Fan et al., 2024) which seek to learn better prior GP hyperparameters such as length-scales and kernel amplitudes based on offline pretraining or manually designed feature representations for combinatorial objects (Deshwal et al., 2023; White et al., 2021; Ru et al., 2021), while keeping underlying kernel definitions fixed.

Numerous end-to-end neural network-based approaches such as the use of attention mechanisms and Transformers (Vaswani et al., 2017) have been introduced to allow more learnable behaviors, and we refer the reader to (Song et al., 2024b) which provides a general reference on their use for black-box optimization. Relevant to our particular case of regression, works such as (Nguyen & Grover, 2022; Garg et al., 2022) demonstrated the benefits of using raw Transformers as in-context learning (ICL) based regression models, or *Neural Processes*, with others (Bai et al., 2023; Zhang et al., 2024) established provable guarantees, and (Müller et al., 2022) demonstrated that Transformers trained on synthetic data as "prior-fitted networks" are capable of Bayesian inference, leading to their use in Bayesian optimization (Müller et al., 2023; Nguyen et al., 2023; Nguyen & Grover, 2024).

Unfortunately, as both raw Transformers and GPs require fixed dimensional features, this limits their applications to inputs expressible as tabular features for e.g. hyperparameter tuning, or task-specific embeddings for e.g. chemistry (Maus et al., 2022). Their inherent bias towards smooth functions may also be suboptimal for certain regression tasks (Grinsztajn et al., 2022). Further works have attempted to improve the flexibility of regression-modeling through the use of token-based representations, which allows regressors to be simultaneously used over various types of input formats. Since the context-window of Transformers still remains the most expensive limitation, a useful organization of recent works can be based on their treatment of the total sequence length, roughly equal to:

$$\text{(number of trials)} \times \text{(average trial token length)} \tag{1}$$

Chen et al. (2022) use custom tokenizations to minimize trial token length in order to maximize trial count. However, this is restricted to very constrained search spaces (e.g. flat hyperparameter spaces), and lacks flexibility in utilizing arbitrary forms of data. Liu et al. (2024); Vacareanu et al. (2024) use text-to-text chat-based services such as ChatGPT (OpenAI, 2022) and Gemini (Google, 2024) to demonstrate their emergent capabilities for ICL regression, but such methods lack the ability to pretrain over large amounts of offline evaluations, and the use of lengthy natural language can lead to limited trial count. In contrast, Song et al. (2024a); Akhauri et al. (2025) avoid ICL altogether and only places a single trial in the context window to allow arbitrary string representations per trial, but requires the use of alternative but tedious methods of absorbing online data, such as inference-time fine-tuning.

In relation to broad literature such as in LLM reasoning, inference-time compute methods can also be seen through the lens of Bayesian Optimization. For example, verifiers such as Process Reward Models (Lightman et al., 2024) can be seen as ICL-based regression methods over chains of thought, but have only used coarse

discretized scores of e.g. $\{-1, 0, 1\}$ rather than highly precise numeric predictions or uncertainties over vastly different scales.

For optimization, we require a regressor with all of the following capabilities:

- Pretrainable over offline evaluations to allow effective meta-learning.
- Flexible representation of inputs with raw strings for application in multiple domains.
- Allow long-range in-context regression using multiple previous evaluations.
- Perform highly precise numeric predictions and uncertainty quantification over diverse objective scales.

This naturally leads to the use of embedding-based methods which can compress any string representation of a candidate into a feature vector, using only a single unit of sequence length when sent to a ICL model such as a raw Transformer. While recent works have applied embeddings for chemical reactions (Kristiadi et al., 2024; Ranković & Schwaller, 2023) and prompt optimization (Hu et al., 2024), no work has assessed their use in the most competitive and widely studied field of standard black-box optimization over tabular-like search spaces, despite evidence (Tang et al., 2025) demonstrating LLM embeddings to possess promising traits such as Lipschitz continuity over tabular features.

## 3 Method

### 3.1 Preliminaries

Let $f : \mathcal{X} \to \mathbb{R}$ be a real-valued function over a search space $\mathcal{X}$. The goal of black-box optimization is to find an input $x^*$ which maximizes $f$:

$$x^* = \arg\max_{x \in \mathcal{X}} f(x) \tag{2}$$

A regressor is a predictive model that estimates a distribution over possible values of $f(\cdot)$ at a given query point $x$, based on the observed history $\{x_s, y_s\}_{s=1}^{t}$ from $t$ previous. Such regressors may also be *learnable* over additional offline data in addition to the given history.

During inference, the regressor can be transformed into an acquisition function $a : \mathcal{X} \to \mathbb{R}$ to guide explore-exploit tradeoffs. We assume the existence of a (potentially task-dependent) *acquisition optimizer* which can quickly and cheaply sample suggestions $x \in \mathcal{X}$, usually in the form of an evolutionary algorithm. The history-dependent acquisition $a_{t+1}(\cdot)$ may then be used to filter out poor candidates or used in an entire Bayesian optimization loop, in which the optimizer identifies the next proposed solution as $x_{t+1} := \arg\max_{x \in \mathcal{X}} a_{t+1}(x)$.

Below in Sections 3.2 and 3.3, we use standard Bayesian optimization modeling techniques as found from previous literature, in particular Transformer Neural Processes (Nguyen & Grover, 2022) and Google Vizier's (Golovin et al., 2017) Gaussian Process Bandit algorithm (Song et al., 2024c). We emphasize that these specific choices are not the purpose of this work and can be interchangeable with other reasonable regression bodies (Müller et al., 2023) or acquisition functions and optimizers (Balandat et al., 2020b) - the reader is free to skip such details.

### 3.2 In-context Transformer Regressor

An embedding-based regressor uses an *embedder* $\phi : \mathcal{X} \to \mathbb{R}^d$ to map a suggestion $x$ to a fixed-length representation $\overline{x} \in \mathbb{R}^d$, which can then be used as a regular feature vector for a numeric regression model. A *string-based* embedder first represents $x$ as a string, which is then passed to a language model for embedding. We specifically use the typical definition of language model embedding, in which we apply a forward pass of the underlying model (encoder or decoder) on the (tokenized) string representation to obtain all token logits in $\mathbb{R}^{L \times d}$, and then average-pool across the length axis to obtain a vector in $\mathbb{R}^d$. We discuss specific string representations in our experiments in Section 4.

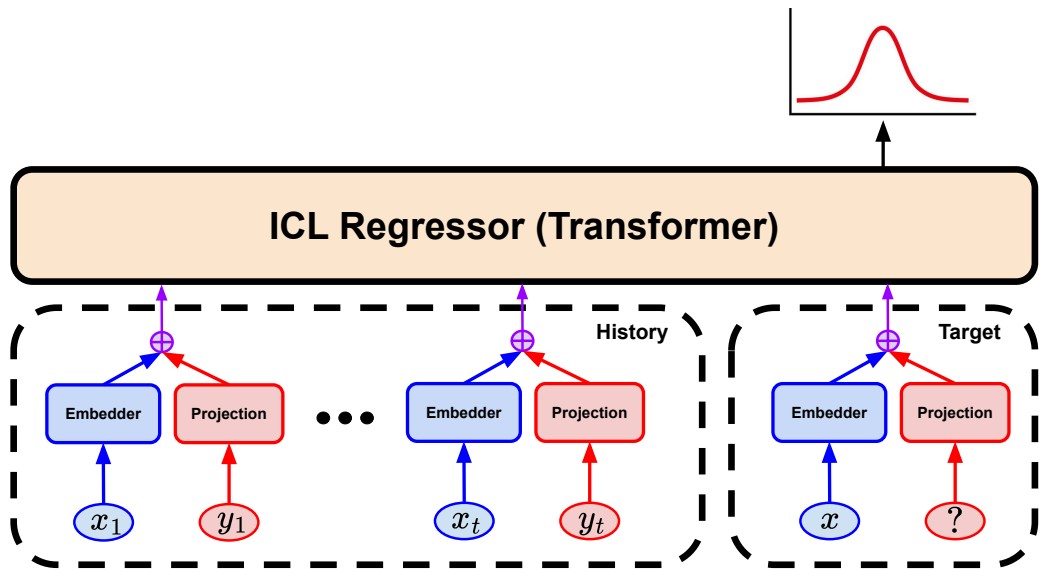

Figure 2: Overview of our model. Most notably, candidates $x$ are converted to language model embeddings to be ultimately used as fixed dimensional features.

For our underlying regression model, we use a standard Transformer Neural Process (Nguyen & Grover, 2022), which takes in as input sequence $(\overline{x}_1 \oplus \overline{y}_1), \ldots, (\overline{x}_t \oplus \overline{y}_t)$ where $\overline{y} \in \mathbb{R}^d$ is the feature representation of the float $y$ after applying a trainable projection, and $\overline{x} \oplus \overline{y}$ is the trial representation expressed as the concatenation of $\overline{x}$ and $\overline{y}$.

To generate a prediction for a query point $x$, we append the query representation $(\overline{x} \oplus \overline{0})$ to the input sequence where $\overline{0}$ is a dummy value. During the forward pass, the Transformer processes this augmented sequence, allowing the query to attend to all previous trials. The model then produces a parametric output distribution over $\mathbb{R}$ at position $t+1$. Specifically, we assume a Gaussian distribution $\mathcal{N}(\mu_{t+1}(x), \sigma_{t+1}^2(x))$, where the mean and standard deviation are predicted by dedicated output heads. Figure 2 provides an overview of the key components of our method.

Additional techniques below are optional but may stabilize training and prediction:

**Parallel Predictions:** In order to simultaneously predict over a given set of $k$ *target* points $x_{t+1}, \ldots, x_{t+k}$, we additionally append $(\overline{x}_{t+1} \oplus \overline{0}), \ldots, (\overline{x}_{t+k} \oplus \overline{0})$ to the history sequence and generate a custom attention pattern of shape $(t+k) \times t$ where all tokens attend to the history while no tokens attend to the targets. This allows an efficient parallel modeling of $p(y_{t+i} \mid x_{t+i}, \{x_s, y_s\}_{s=0}^t)$ for $1 \le i \le k$ which speeds up training when computing a summed loss over multiple target predictions.

**$y$-Normalization:** Depending on the function, $y$-value outputs may contain a wide variety of scales, and need to be normalized properly. We use the same normalization procedure as in Google Vizier (Song et al., 2024c), modified to allow incoming target values. These steps consist of, in order: (1) Shifting objectives to have zero mean and divide by standard deviation. (2) Reducing harmful effects of bad outliers by fitting the "bad half" of objectives $\{y_i \le y_{\text{median}}\}$ to a normal curve, using percentiles as z-scores. (3) Linearly scale $y \leftarrow \frac{y - y_{\min}}{y_{\max} - y_{\min}}$ which ensures all historical $y$-values within $[0, 1]$ and apply additional damping (e.g. sigmoid or log transform) to target values significantly outside this range.

**Encoding Metadata:** Many times there may be a *metadata $m$* associated to an objective $f$, which provides useful prediction information on the behavior of $f$ or simply can inform the model of a new objective or search space. We can also embed this metadata as an additional feature $\overline{m}$, which is concatenated to every $\overline{x}$ similarly to standard encoder-decoder techniques (Raffel et al., 2020).

### 3.3 Pretraining and Inference

Denote a task $\mathcal{T} = (f, \mathcal{X})$ as a specific objective function over a particular search space.

**Pretraining:** We assume a collection of offline *training tasks* $\{\mathcal{T}_1, \mathcal{T}_2, \ldots\}$, with different search spaces and objective functions, with each task containing its own collection of evaluated trials $\{x_s, y_s\}_{s=1}^T$ where $T$ is the (potentially task-specific) offline trajectory length.

While the embedder is frozen, we pretrain the weights $\theta$ of the ICL regression Transformer, over all such offline evaluation data. Each training example consists of a sampled task and history cutoff length $t' \in [0, T)$ so that $\{x_s, y_s\}_{s \leq t'}$ is considered a history, while $\{x_{t'+i}, y_{t'+i}\}_{t'+i}^T$ are target points, with the loss computed as the mean prediction loss over all targets, i.e.

$$\frac{1}{T - t'} \sum_{i=1}^{T-t'} \ell_\theta(x_{t'+i}, y_{t'+i}; \{x_s, y_s\}_{s=1}^{t'}) \tag{3}$$

where $\ell_\theta(x, y; \{x_s, y_s\}_{s=1}^t)$ is the negative log-likelihood using our Gaussian output distribution, of predicting $y$ given $x$ and history $\{x_s, y_s\}_{s=1}^t$.

**Inference:** At inference, we use our mean and deviation head to form a UCB-based acquisition $a_{t+1}(x) = \mu_{t+1}(x) + \sqrt{\beta} \cdot \sigma_{t+1}(x)$ where $\sqrt{\beta}$ is a problem-dependent constant. We use a (potentially domain-dependent) zeroth-order optimizer such as evolutionary search to maximize this acquisition, and thus only require forward passes, although gradient-based acquisition maximization is possible with soft-prompt optimization techniques (Lester et al., 2021).

Since there may be distributional shifts for parameter names encountered between pretraining and inference, we may either apply data augmentation by randomizing parameter names during pretraining, or transform the search space during inference to match those encountered in pretraining.

### 3.4 Model Details

In this paper, to demonstrate the validity of our approach on relatively low compute budgets, we intentionally use relatively smaller language model embedder sizes in comparison to the larger and significantly more expensive GPT (OpenAI, 2023) or Gemini (Google, 2024) family of models. Specifically, we use a pretrained T5-XL encoder (1B parameters), based on the encoder-decoder T5-family of models (Raffel et al., 2020). Along with only 8 layers of the underlying regression Transformer, this leads to a maximum required training budget of approximately 16 GPUs for training and 1 GPU for inference, possible with most academic budgets. Further details with respect to model sizes and training hyperparameters can be found in Appendix A.

The cheap inference cost is also necessary when the acquisition function may be called thousands of times by a zeroth-order acquisition optimizer per candidate proposal. It is worth noting that time and memory complexity costs may even further be reduced using efficient Transformers (Tay et al., 2022). Faster embedders lead to large constant factor reductions, while faster regressors can lead to linear $\widetilde{O}(t)$ complexities with respect to the number of trials.

## 4 Experiments

### 4.1 End-to-End Black-box Optimization

We focus on demonstrating the general applicability of LLM embeddings across a variety of tasks rather than achieving the best possible result against very domain-specific baselines, although we outline possible improvements within specific domains in Section 5. In tabular settings, we purposely use the same algorithmic components (other than replacing the GP regressor) as Google Vizier's UCB-based "GP-Bandit" (Song et al., 2024c), which allows direct apples-to-apples comparisons. We stress the importance of clean comparison, as it is well known that other components in the Bayesian Optimization pipeline (e.g. choice of acquisition and acquisition optimizer) also strongly affect performance and can lead to confounding factors. GP-Bandit was also reported to be near-optimal across well-known five different industry methods such as Optuna (Akiba

et al., 2019) and Ax (Balandat et al., 2020a), and thus achieving competitive results will demonstrate the easy integration of embedding-based regressors into state-of-the-art industry-grade optimization pipelines.

We evaluate the performance of our algorithm on various problems consisting of synthetic, combinatorial, and hyperparameter optimization (exact details in Appendix B). All curves are plotted with mean and 0.5 deviation as error bars over 10 repeats.

**Synthetic Optimization:** In common optimization scenarios, the search space is a flat Cartesian product of float and categorical parameter types. Our string-based regression will represent each $x$ with standard JSON over the dictionary mapping parameter names to values, e.g. for a search space with two parameters, one continuous named `p0` and another categorical `p1`, the string representation for an example trial would be `{"p0":0.3,"p1":"category_1"}`.

We benchmark over the Black-box Optimization Benchmarking (BBOB) suite (ElHara et al., 2019), one of the most widely used synthetic function benchmarks, containing 24 different objectives over continuous search spaces. In order to have a notion of offline "training" data and unseen "test" functions to be optimized, we split the original functions across each landscape type (Separable, Ill-Conditioned, etc.) into training and test sets, and additionally apply randomized transformations (e.g. shifting, rotating, discretizing, increasing/decreasing dimensions) over all objectives to induce non-continuous search spaces with categorical parameters and avoid overfitting. Evaluations used for offline training data were uniformly sampled over the search space.

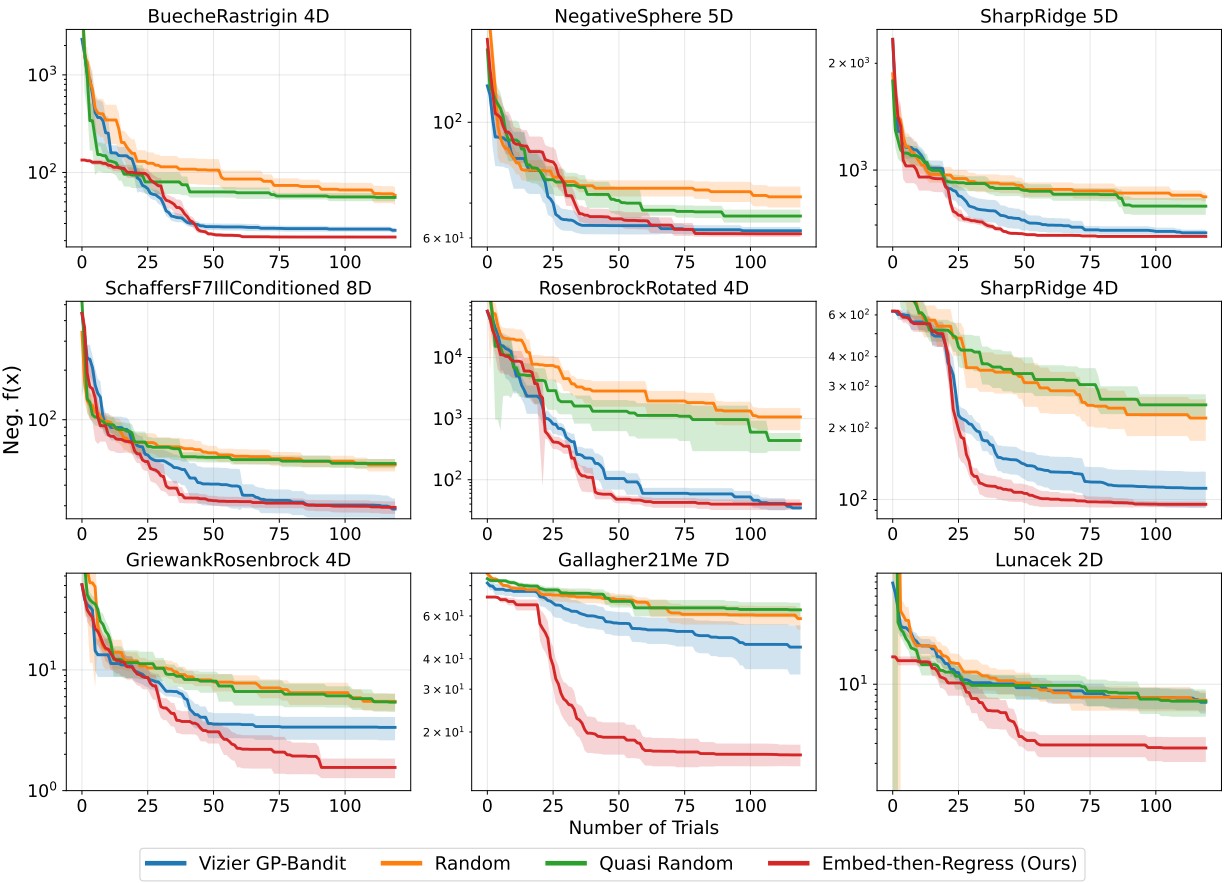

Figure 3: (↓) Lower is better. Mean optimality gap curves across 9 randomized test functions, some with non-continuous parameters. Note: $y$-axis is log-scaled to depict clearer separation between baselines.

In Figure 3, we find that Embed-then-Regress is generally comparable with and interestingly can even significantly outperform GP-Bandit in a few cases.

**Combinatorial Optimization:** We further benchmark over combinatorial objectives whose search spaces are typically difficult to regress over. Many of these can be found in common operations research literature, e.g. permutation-based (Travelling Salesman, Quadratic Assignment, Flowshop Scheduling, and N-Queens), and choice-based (submodular maximization problems such as covering and log-determinant functions).

Each of these problems can be parameterized by a set of coefficients (e.g. city locations for Travelling Salesman, matrix entries for log-determinant). Note that we are in the *bandit* setting, in which these coefficients are hidden from the algorithm and the only feedback is the final objective. Similar to before, we thus can also generate offline pretraining data by randomizing these coefficients and problem sizes, and evaluating over random candidates. For our string regression, we may simply use JSON over indices; e.g. `{[0]:2, [1]:0, [2]:3, [3]:1}` for a permutation space of size 4, e.g. `{[0]:1, [1]:3}` for a $\binom{4}{2}$ choice space.

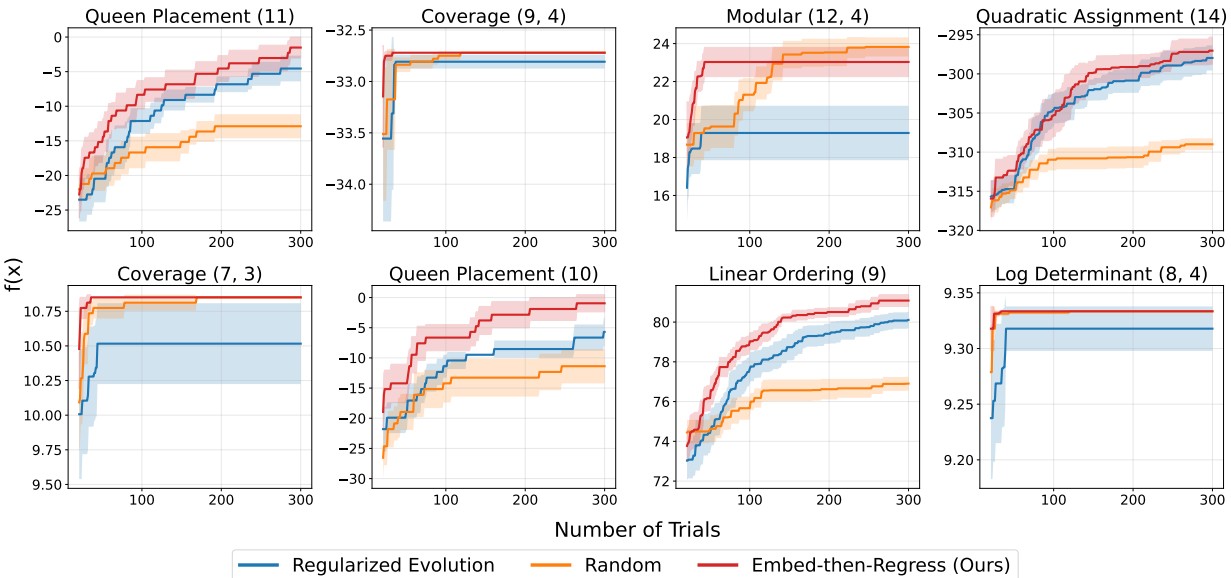

Figure 4: (↑) Higher is better. Best-so-far curves across 8 randomized combinatorial problems. Title parenthesis ($P$) means a permutation space of size $P$ and $(N, K)$ denotes a $\binom{N}{K}$ choice space. Note that plotting begins at trial 20, since previous trials are random.

While there are few previous works using GPs for e.g. permutation spaces (Deshwal et al., 2022; Oh et al., 2022), they require constructing very domain-specific kernels and complex acquisition optimizers (e.g. semi-definite programming) making them difficult to reproduce. We thus use a simpler optimizer such as Regularized Evolution (Real et al., 2019) which does not need modeling assumptions other than implementing random mutations between trials and can be used broadly (Real et al., 2020). We also empirically found this was better than other evolutionary alternatives such as NSGA-II (Deb et al., 2002) or hill-climbing.

Rather than fully optimizing the acquisition which can be slow for large-scale evolutionary searches, we can simply apply best-of-many sampling by using the regressor's UCB acquisition to rank sample candidates proposed by evolution, and suggest only the best. In Figure 4, we see that this boosts exploration over the original Regularized Evolution, which can often get stuck at local optima early on.

**Hyperparameter Optimization:** With the advent of industry-wide hyperparameter tuning services (Golovin et al., 2017; Balandat et al., 2020b; Liaw et al., 2018), large amounts of evaluations over expensive but realistic experiments can be stored offline. To efficiently benchmark our method over objectives encountered in the wild, we use surrogate-based benchmarking which has shown to be effective (Zela et al., 2022; Eggensperger et al., 2015) for providing realistic yet cheap comparisons between different algorithms without using huge amounts of compute. Without surrogates, benchmarking could take weeks in wall-clock time and hundreds to thousands of GPU-hours.

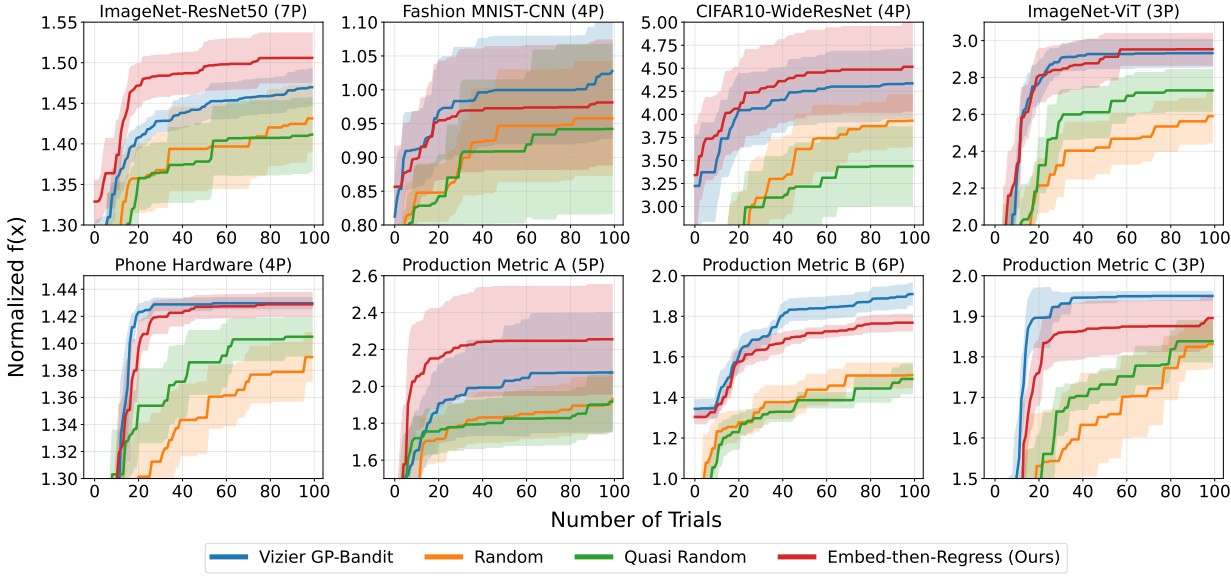

Figure 5: (↑) Higher is better. Best-so-far curves over 8 randomly chosen hyperparameter surrogate functions. Title contains task summary along with number of parameters (#$P$). Normalized objective values are displayed since raw objective values over large ranges, e.g. $y \in [-10^7, 10^7]$ from private functions would lack meaningful context.

To avoid advantaging either GP-Bandit or our neural network regressor during optimization, we apply a tree-based surrogate via XGBoost (Chen & Guestrin, 2016) for each study's offline evaluations and use the surrogate's interpolation abilities to evaluate new candidates and act as a realistic online objective function. We select 20 representative and diverse hyperparameter tuning tasks. These include image classification, hardware, and also production metrics.

While it is possible to train our regressor on top of expensive evaluations similarly to (Chen et al., 2022), this is typically impractical for most academic research settings due to the lack of such data. To demonstrate the wide applicability of our method without requiring expensive training data, we use the same regressor model trained only on synthetic BBOB trajectories as found previously in Figure 3.

In Figure 5, we display randomly chosen objective functions and see again that Embed-then-Regress is generally comparable against GP-Bandit. This is corroborated when we aggregate performances over all tasks in Figure 6, where we see our method achieves the same median log-efficiency (defined in Appendix B.3) as GP-Bandit. This suggests that the BBOB-trained model has learned a general regression capability which can be transferred to very different unseen tasks at inference time.

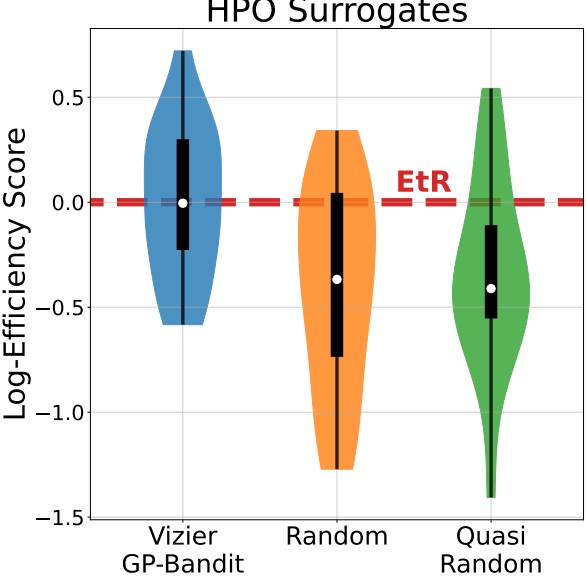

Figure 6: (↑) Higher is better. Violin plots of log-efficiency scores across algorithms over 20 hyperparameter optimization (HPO) surrogate objectives. Embed-then-Regress (EtR) displayed horizontally at Log-Efficiency=0 as reference.

### 4.2 Ablations

In this subsection, we ablate different effects on the model's prediction ability, which directly affects optimization performance. We perform comparisons using a variety of predictive metrics, consisting of

negative log-likelihood (NLL), mean average error (MAE), R-Squared, and mean absolute calibration error (MACE) (Chung et al., 2021).

**String Embedder Size:** In Figure 7, we see that the size of the pretrained string embedder has a monotonic influence on the predictive performance over BBOB evaluations. As we vary the T5 embedder sizes (Small, Large, XL), there is a clear trend across all predictive metrics computed over normalized $y$-values.

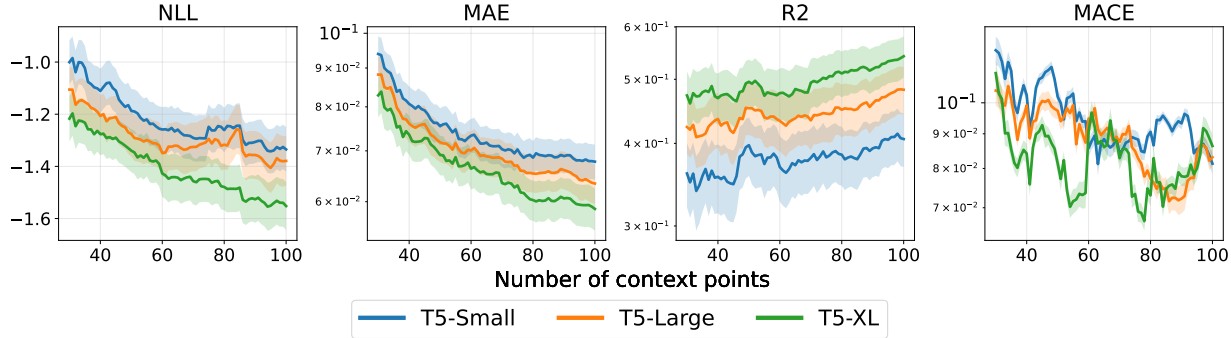

Figure 7: $(\downarrow, \downarrow, \uparrow, \downarrow)$ are better, respectively. Number of historical context points vs predictive metrics on unseen points over unseen BBOB function trajectories, while varying string embedder sizes. Solid line denotes mean over 10 test functions and error bars denote standard deviation.

It is interesting to note that larger encoders, which are pretrained over mostly English text, lead to better predictive performance over BBOB representations which do not contain any English words. Considering that the embedder's weights are also frozen, this trend potentially suggests that larger language models inherently provide better features even for numeric data formats.

**ICL Transformer Size:** In Figure 8, we find that the ICL Transformer size also plays a role, where higher layer sizes lead to better predictive outcomes. In contrast to the string embedder, here the ICL model's weights are trainable, and thus larger models can potentially possess higher capacities and better inductive biases to train over the same offline data.

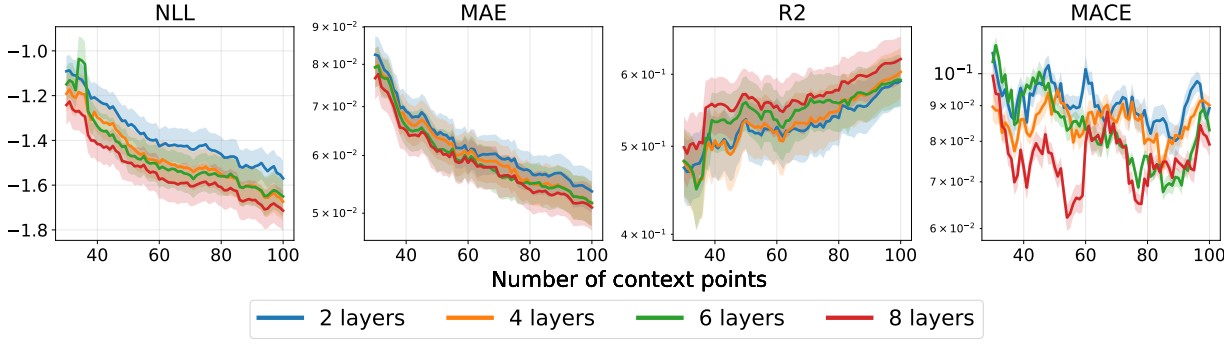

Figure 8: Analogous setting to Figure 7, while varying the number of attention layers of the ICL Transformer.

Overall in both cases for Figures 7 and 8, we verify in-context regression occurring for different test functions, where more context points leads to better predictive performance.

## 5 Conclusion and Future Work

We investigated the use of string-based ICL regression for Bayesian Optimization over a variety of synthetic, combinatorial, and real-world optimization problems, and shown it to obtain comparable results against industry-standard GP baselines, yet allow flexibility in esoteric spaces such as permutations and combinations.

We leave architectural improvements for future work - for example, the aggregation method over Transformer outputs may be learned rather than predefined using average pooling as in this paper, and more broadly there may be better methods for computing fixed-length embeddings from a Transformer. Further investigation into different regression bodies could lead to e.g. string-based GPs using kernels over string embeddings for better guarantees of uncertainty estimates.

As strings are significantly more flexible representation formats of different data types, an ambitious and exciting direction is to pretrain a unified in-context regression model broadly over multiple different domains including prompt optimization (Fernando et al., 2024) and code search (Romera-Paredes et al., 2023), in order to obtain a "universal" in-context regressor which can speed up search over evolutionary algorithms. Additionally, outside of optimization problems which are stateless with respect to inputs, it is worth investigating whether such methods are applicable for LLM reasoning as reward models to assist tree search-based approaches (Yao et al., 2023) for stateful environments in language modeling.

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

# Appendix

## A  Model Details

The full list of hyperparameters:

- ICL Transformer Size: 1024 feature dimension, feedforward projection outputs of 4096, and 8 layers of multi-headed attention with 16 heads.

- String-Embedding: We use the T5-XL encoder. Strings were clipped to a maximum of 400 tokens, using the SentencePiece tokenizer (Kudo & Richardson, 2018) with a vocabulary of 32000 subword tokens.

- Training: Effective batch size of 16, learning rate of $5 \times 10^{-4}$, weight decay of $10^{-5}$, gradient clipping of 0.5, using the AdamW optimizer. A fixed number $T \geq 100$ total trials were always placed in the context window, with the number of history trials $t'$ sampled between $[10, T-10]$ and the rest were target points for loss computations.

- Inference: UCB coefficient $\sqrt{\beta} = 1.8$.

For traditional (synthetic and hyperparameter) optimization tasks, we used a maximum budget of 1000 evaluations for the Firefly acquisition optimizer for both GP-Bandit and our method.

For combinatorial tasks, Regularized Evolution used a population size of 50, and a tournament size of $7 \approx \sqrt{\text{population size}}$, as prescribed in (Real et al., 2019). When it was augmented by the acquisition, we chose the highest scoring proposal out of 5 samples as the final candidate for evaluation. The acquisition-based ranking began at trial 20.

## B  Benchmarking

For every algorithm and objective pair, we run 20 seeds and plot the best-so-far mean with $0.5 *$ (standard deviation) as error bars.

### B.1  BBOB

Our train-test split is performed equally across all landscape types[1] (separable, low/moderate condiitoning, high conditioning + unimodal, multi-modal with global structure, multi-modal with weak global structure):

- Train: {Sphere, Ellipsoidal, Rastrigin, AttractiveSector, StepEllipsoidal, Ellipsoidal, Discus, BentCigar, Weierstrass, Schwefel, Gallagher101Me}

- Test: {BuecheRastrigin, LinearSlope, RosenbrockRotated, SharpRidge, DifferentPowers, SchaffersF7, SchaffersF7IllConditioned, GriewankRosenbrock, Gallagher21Me, Katsuura, Lunacek, NegativeSphere, NegativeMinDifference, FonsecaFleming}

For transformations, we applied the following, given an initial function $f : \mathbb{R}^{\text{dim}} \to \mathbb{R}$:

- Shifting: Uniformly samples a shift $c \in \mathbb{R}^{\text{dim}}$, and transform $f(x)$ into $f(x - c)$.

- Rotation: Uniformly samples an orthonormal matrix $\mathbf{R} \in \mathbb{R}^{\text{dim} \times \text{dim}}$ and transforms $f(x)$ into $f(\mathbf{R}x)$.

- Discretization: Each parameter is randomly chosen to remain continuous, or either a `DISCRETE` or `CATEGORICAL` parameter, whose feasible points are selected over a uniform grid between the original bounds $[-5, 5]$, with the number of feasible points uniformly selected from 2 to 16.

The offline training dataset consisted of 1M tasks over sampled training objectives (along with random transformations) with each trial randomly sampled from its corresponding search space.

---

[1] https://numbbo.github.io/coco/testsuites/bbob

## B.2 Combinatorial

We implemented both objective functions and evolutionary algorithms in PyGlove (Peng et al., 2020), a framework for evolutionary and combinatorial optimization.

**Permutation:** Let $x$ be a permutation of $[n] = \{1, 2, \ldots, n\}$ where $x^{(i)}$ denotes the permutation index at position $i$.

- Travelling Salesman: $f(x) = -\sum_{i=1}^{n-1} \|\text{City}(x^{(i)}) - \text{City}(x^{(i+1)})\|_2$ where each city's location is randomly sampled from $\mathbb{R}^2$

- Flowshop Scheduling: $f(x) = -\sum_{i=1}^{n} C_{i,x^{(i)}}$ where $C \in \mathbb{R}^{n \times n}$ is a random set of costs.

- Linear Ordering: $f(x)$ is the upper-triangular sum of the corresponding matrix after applying a permutation of rows and columns on $W \in \mathbb{R}^{n \times n}$ using $x$.

- Quadratic Assignment: $f(x) = -\text{Trace}(WPDP^\top)$ where $W, D \in \mathbb{R}^{n \times n}$ are random weight and distance matrices, respectively, and $P$ is the permutation matrix associated with $x$.

- N-Queens: A generalization of the classic 8-Queens problem, in which the $i$-th queen is placed on $(i, x^{(i)})$ and $f(x)$ is negative of the number of pairs of queens which diagonally attack each other.

**Choices:** Let $\text{Ind}_k$ denote the collection of all $k$-sized subsets of $[n]$. We may represent $x \in \text{Ind}_k$ as a set of $k$ indices.

- Modular Function: $f(x) = \sum_{i \in x} w^{(i)}$ where $(w^{(1)}, \ldots, w^{(n)}) \in \mathbb{R}^n$ are random weights.

- Coverage Function: Let $E_1, \ldots, E_n$ be random covers, i.e. subsets of $[n]$ and $(w^{(1)}, \ldots, w^{(n)}) \in \mathbb{R}^n$ be random weights. Let $\text{UnionCover}(x) = |\cup_{i \in x} E_i|$. Then $f(x) = \sum_{j \in \text{UnionCover}(x)} w^{(j)}$

- Log Determinant: Given a randomly sampled positive semi-definite matrix $M \in \mathbb{R}^{n \times n}$, $f(x) = \log \det(M')$ where $M' \in \mathbb{R}^{k \times k}$ is the minor of $M$ using the indices from $x$.

Offline data collection was done similarly to BBOB (i.e. random problem with random trial sampling), to generate 1M tasks and corresponding trajectories.

## B.3 Hyperparameter Optimization

In order to saturate the search space with real trials for more accurate surrogates, each offline hyperparameter optimization task was ensured to have at least 100 offline trials.

For tree-based surrogates, we use the standard API (`XGBRegressor`, `XGBRFRegressor`)[2] found in XGBoost, with typical grid-search hyperparameter sweeps for each study:

- `"min_child_weight"`: $[1, 5, 10]$

- `"learning_rate"`: $[0.001, 0.01, 0.1]$

- `"gamma"`: $[0.0, 0.3, 0.5]$

- `"subsample"`: $[0.6, 0.8, 1.0]$

- `"colsample_bytree"`: $[0.6, 0.8, 1.0]$

- `"max_depth"`: $[3, 5, 7]$

---

[2]https://github.com/dmlc/xgboost

In Figure 6 in the main body, to aggregate performances over objectives of different scales and landscapes, for each function $f$ and baseline algorithm $\mathcal{A}$, we compute the log-efficiency metric introduced in (Song et al., 2024c), as it is scale-independent. Roughly speaking, a baseline algorithm with a log-efficiency of $c$ requires a factor of $\exp(-c)$ trials to reach the same level of performance as the reference $\mathcal{A}_{\text{ref}}$, which is our proposed Embed-then-Regress method.

More formally, we can define

$$\text{LogEfficiency}(y \mid f, \mathcal{A}, \mathcal{A}_{\text{ref}}) = \log\left(\frac{\text{RequiredBudget}(y \mid f, \mathcal{A})}{\text{RequiredBudget}(y \mid f, \mathcal{A}_{\text{ref}})}\right) \tag{4}$$

where $\text{RequiredBudget}(y \mid f, \mathcal{A})$ is the minimum trial budget required by $\mathcal{A}$ to reach $y$ on $f(\cdot)$, and our final log-efficiency score is computed over the median of individual log-efficiencies when $y$ varies over the averaged best-so-far curve between $\mathcal{A}$ and $\mathcal{A}_{\text{ref}}$.

## C   Example String Representations

Below, we provide some string representations of inputs $x$ from different optimization tasks. While our experiments in the main body did not use metadata $m$, we provide examples for when $m$ may be used to distinguish between different tasks.

| Benchmark | Example $x$ | Example $m$ (if applicable) |
|---|---|---|
| Traditional | `x0:-0.3`
`x1:4.5`
`x2:-1.2`
`x3:-4.1` | `x0:DOUBLE,[-5,5]`
`x1:DOUBLE,[-5,5]`
`x2:DOUBLE,[-5,5]`
`x3:DOUBLE,[-5,5]` |
| Combinatorial (Permutation) | `[0]: 0`
`[1]: 4`
`[2]: 2`
`[3]: 1`
`[4]: 3` | `task:"Permutation"`
`size:4` |
| Combinatorial (Choice) | `[0]: 1`
`[1]: 3` | `task:"Choice"`
`size: 4-choose-2` |

