# OpenReview forum: "Language Model Embeddings Can Be Sufficient for Bayesian Optimization"
_TMLR — Rejected by TMLR_

### Review · Reviewer_shXW · 2025-08-08

**Summary Of Contributions:**

This paper proposes a new LLM-based paradigm for Bayesian optimization.
By using an LLM as an embedder, it can handle variable-length inputs.
The authors also demonstrate, on both industrial and synthetic datasets, that their algorithm outperforms other approaches.

**Audience:**

Yes

**Audience Explanation:**

Bayesian regression is a popular research topic and is widely applied in real-world scenarios.

**Claims And Evidence:**

Yes

**Claims Explanation:**

The paper conducts extensive experiments to validate its claims.

**Requested Changes:**

I have the following questions and suggestions:

1. “Unlike tabular models, LLMs are not constrained by fixed-length input representations, making them more adaptable across different tasks.”
I believe that a standard Transformer can also address variable-length inputs by using masking, so why is it necessary to use LLMs here? Typically, LLMs excel in text-based tasks but do not have a clear advantage for purely numeric tasks. Please refer to this paper, which recommends using tree-based models for such cases.

2. “Our framework ‘embed-then-regress’ describes a simple recipe for using free-form string representations in Bayesian Optimization, by using language models to embed trial inputs as features for in-context regression.”
Could you clarify why LLMs are preferred over lighter embedding methods such as GloVe? Using GloVe embeddings would be far more computationally efficient. Could you add an ablation experiment comparing LLM embeddings with GloVe embeddings?

3. For "Parallel Predictions: In order to simultaneously...history sequence", it seems that you missed an $\oplus$.

4. In Figure 4, why does the “random” baseline achieve such strong performance in some subfigures? Also, why is the number of trials for “random” smaller in certain subfigures?

---

> ### Author Response · Authors · 2025-09-02
> **Response to Reviewer shXW**
>
> Thanks for the positive feedback and recognizing our contributions! We appreciate the thoughtful questions and provide our responses below.
>
> 1. **"why is it necessary to use LLMs here?"** - To clarify our motivation and the distinct roles of the model components:
>     * **Role of LLM:** The pretrained LLM is used only as a feature extractor to embed a structured string (JSON) representation of parameters $x$ into a meaningful, fixed-length vector. The subsequent task of regressing to a numeric objective value is handled by a separate Transformer (i.e. Neural Process) trained from scratch on these embeddings. This separation of roles leverages the strengths of each component effectively.
>     * **Representation Flexibility:** While a standard Transformer can indeed handle variable-length token sequences, our contribution is more about the flexibility of the input representation itself. Prior work [1] that used Transformers for Bayesian Optimization required highly customized tokenizers and was often restricted to domains with flat hyperparameter spaces. Our method, by accepting any search space that can be described as a free-form string (like JSON), is more general and readily applicable across diverse and structured domains such as combinatorics.
>     * **Reference to Tree-Based Models:** Thanks for the suggestion, but the link seems to be missing. We'd be very interested in reading it if you could provide the reference.
>
> 2. **"GloVe embeddings"** - We chose a pretrained LLM embedder over methods like GloVe for two main reasons:
>     * **Representation quality:** LLMs like T5 are pretrained on vast and diverse corpora that include not just natural language, but also code and structured data. This gives them a better understanding of structured formats like the JSON strings (
> `{parameter: value}`) that we use. This also leads to more semantically meaningful embeddings that capture the relationships between parameters better than GloVe, which was trained primarily on natural language text.
>     * **Computational efficiency:** While GloVe is certainly lighter,  the computational overhead of our chosen embedder, T5-XL encoder (1B parameters), is minimal in practice. Since T5-XL is also open-sourced, it can be deployed locally with very low latency for the single forward pass needed for embedding. While we are not able to perform a full ablation with traditional embedding methods like GloVe during the rebuttal period, we believe it is an interesting direction for future work.
>
> 3. **Typo** - fixed.
>
> 4. **Figure 4, combinatorial optimization results:**
>     * **Random baseline performance:** For combinatorial landscapes, heuristic search methods like Regularized Evolution can sometimes get stuck in a local optimum. Pure random search, on the other hand, continually explores the entire search space and is not susceptible to getting stuck.
>     * **Number of Trials:** All methods, including random search, were run for the full 300 trials. The plotted lines for the random baseline may appear shorter in some subfigures because its best-so-far performance converges or plateaus early on and overlaps with other methods.
>
> [1] Towards Learning Universal Hyperparameter Optimizers with Transformers

---

> > ### Comment · Reviewer_shXW · 2025-09-03
> >
> > The paper I mentioned is "Why do tree-based models still outperform deep learning on tabular data?". Please refer to it.

---

> > > ### Author Response · Authors · 2025-09-16
> > > **Added Reference**
> > >
> > > Thanks for the useful reference. We've added it in Section 2 (in blue), where it also shows that raw tensor-based Transformers and GPs are inherently biased towards smooth functions, which is not necessarily optimal for certain regression tasks.
> > >
> > > In contrast, token-based representations do not have such biases.

---

### Review · Reviewer_Tkcu · 2025-08-23

**Summary Of Contributions:**

This paper focuses on the regression modeling part of Bayesian Optimization, specifically exploring the use of language model embeddings and transformers to map the string inputs of the regression problem to an objective value. This approach inherits the benefits of pretrained language models and offline pretraining. Experiments on synthetic, combinatorial, and hyperparameter optimization tasks show competitive results to standard and industry methods.

Strengths:
1. The method makes sense. Makes it seem weird in hindsight why other people working in this area did not try it as the first approach.

Weaknesses:
1. Bad title. It leads the reader to expect a yes/no answer, which is not provided by this paper. No method in machine learning can be proven to be sufficient for all possible problems for all time.
2. Writing issue: the second statement of contribution does not claim much. "By using a standard...we can achieve uncertainty-aware numeric predictions over unseen objective functions." Any arbitrarily initialized parametric distribution can also "achieve uncertainty-aware predictions", except that it will be unusable. The authors probably meant that the proposed method achieves good/competitive predictions versus their chosen baselines.

**Additional Comments:**

1. Typo on page 7: "which be slow"
2. The emphasis on "fixed-length tensors for input representation" as a limitation of previous methods in the Introduction first paragraph is strange, since your LM embeddings also have a fixed dimension.

**Audience:**

Yes

**Audience Explanation:**

Improving the generality of Bayesian optimization methods with the benefits of LM pretraining would be interesting to a significant fraction of TMRL's audience who work in specialized applications with small datasets.

**Claims And Evidence:**

Yes

**Claims Explanation:**

The claims are supported by evidence:
1. Framework proposal. Clearly described in the method section
2. Experiments in three categories show that the method can make predictions that are competitive with standard methods

**Requested Changes:**

Use a better title, and edit claim 2 to be non-vacuous.

---

> ### Author Response · Authors · 2025-09-02
> **Response to Reviewer Tkcu**
>
> Thank you for the constructive feedback and agreed that such a relatively simple method could've been used earlier!
>
> We address each remaining concern and question below.
>
> 1. **Title:** Agreed that the original question-like title can look misleading. Reworded it to “Language Model Embeddings Can be Sufficient for Bayesian Optimization” which is less of a question, but also doesn't make the aggressive claim to be sufficient all the time.
>
> 2. **Second Contribution:**  We have reworded this contribution to be more precise: "...we can achieve accurate and well-calibrated numeric predictions over unseen objective functions, competitive with specialized, state-of-the-art methods". This more accurately reflects the evidence presented in our experiments.
>
> Additional comments:
> 1. Typo: fixed
> 2. "fixed-length tensors for input representation" changed to "... as their input dimensions are dependent on the problem space".

---

### Review · Reviewer_VPct · 2025-08-27

**Summary Of Contributions:**

The authors propose embeddings from a frozen LLM (T5 models) as input to an ICL Transformer regressor (which is trained) for Bayesian Optimization. They claim that the LLM embeddings offer for a flexible (& esoteric) tasks. The authors show that on some standard benchmarks, their method can reach Google Vizier GP-based method, and do some ablations on the size of the LLM embedder and the depth of the ICL Transformer regressor.

**Audience:**

No

**Audience Explanation:**

In my view, readers of TLMR would be interested in that study, but not in the findings. There is no understanding why the method works, apart from showing that it can match Google Vizier in some cases.

**Claims And Evidence:**

No

**Claims Explanation:**

I am afraid that the claims in the paper are not supported in a proper scientific fashion.

My biggest critique is the usefulness of this approach. The authors claim that LLM-based embeddings are better than other methods since they can accommodate for "esoteric" problems with variable input structures, yet, in all problems the baselines can be used. Furthermore, the benchmarks are quite standard. Is there a clear benchmark where your method can be used, while others not? Is that benchmark meaningful? I didn't get any insight into that.

I didn't get any insight into the types of embeddings that are being created with the LLM. Why LLM-based embeddings as opposed to others? You mentioned Lipschitz-ness of embeddings on tabular data as motivation? How does that (or anything else) manifest in your experiments? I was expecting analysis on the embeddings from the LLM, but this analysis is strikingly lacking.

There is lack of detail on how exactly you construct your embeddings. I gather the approach is to input JSON-like inputs to the LLM, but then are you relying on the fact that the LLM has been pretrained on JSON-like data? Any attribution analysis from the pretraining data (which is public for T5 and it is amenable to study in a TLMR paper) and your data in the paper is missing.

Many remaining questions make this study non-scientific:

How do the randomized transformations induce non-continuous space?

Some of these transformations are continuous, and it's not clear to me what you mean by the effect of the transformations creating a non-continuous space.

Could you explain more about the choices of transformations here?

Why in Lunacek 2D all baselines perform the same as the random baseline, while your method is significantly better?

In many cases in Figure 4, random is better than the Regularized Evolution. Why is that the case? Can you explain the strength of the random baseline.

What are the objective functions on Figure 5?

**Requested Changes:**

* Please use \citet when you cite at the beginning of a sentence;
* to attend to <- attend to.

---

> ### Author Response · Authors · 2025-09-02
> **Response to reviewer VPct, Part 1**
>
> We thank the reviewer for their detailed feedback and the opportunity to clarify several aspects of our work. We address each concern below:
>
> * **"...LLMs...accommodate for "esoteric" problems with variable input structures, yet, in all problems the baselines can be used"**: Please note that to make any claims of comparisons, we’d need to optimize on a domain with solid baselines. If we presented results on domains with no strong baselines, it’d be impossible to tell if our approach is legitimate at all. But also note our key difference against baselines is universal regression, i.e. a single pretrained “Embed-then-Regress” model can be applied directly to diverse optimization tasks with different search spaces without retraining. Our experiments cover both (1) competitiveness and (2) flexibility:
>      * On BBOB and hyperparameter optimization settings (the most competitive arena), we still obtain comparable or superior performance against Vizier GP-Bandit (a SoTA industry-grade optimizer). The model was used for all nine distinct BBOB tasks, which have varying dimensionalities and parameter types.
>      * On combinatorial search spaces, our regressor boosts performance over evolutionary search baselines, while remaining very easy to use via string representations. Other Bayesian Optimization baselines [1, 2] require writing incredibly specific combinatorial kernels, making them notoriously difficult to use.
>
> * **"Insights into the effectiveness of the LLM embeddings over others"**:
>     * A deep analysis of the properties of LLM embeddings for numeric regression is a significant research direction and is worth another paper. In a prior work, Tang et al. [3] provided a thorough analysis, finding that the strong performance of LLM embeddings on numeric data can be partly explained by their inherent preservation of Lipschitz continuity.
>     * Our work is orthogonal but complementary to these findings. We take the effectiveness of these embeddings as a starting point and investigate their practical application: to demonstrate that this powerful embedding space can be leveraged to build a generalizable and competitive method for Bayesian Optimization.
>
> * **Detail of how we constructed our embeddings:**
>     * To construct the embeddings, we first represent each search point as a JSON string mapping parameter names to their values. This string is then passed through the pretrained T5 encoder, and we apply an average pooling operation over the output token representations to get a single, fixed-dimensional embedding vector.
>     * The reviewer is correct that our method implicitly relies on the LLM having been exposed to structured, code-like data during pretraining. Standard T5 models [4] are pretrained on the C4 corpus [5], which includes a wealth of web data containing JSON and other dictionary-like structures. Nearly all modern LLMs are trained on such data, implying our method is broadly applicable.
>
> * **How do the randomized transformations induce non-continuous space**: One of the transformations we apply is the discretization of continuous parameters. For example, a continuous dimension with bounds [−5, 5] is randomly converted into a DISCRETE or CATEGORICAL parameter with a set number of feasible points (e.g., 2 to 16 equally spaced values). This transformation explicitly creates a non-continuous search space. The full details of all transformations used are in Appendix B.1.
>
> * **Why is our method significantly better than other methods in Lunacek 2D?:** One potential explanation is that the Lunacek objective may violate the prior assumptions of the Gaussian Process model, causing it to perform poorly. Our Transformer-based regressor does not have such a strong inductive bias. By learning from a diverse set of offline data, it may have developed a more flexible "prior" that better captures the structure of this particular function, leading to more effective optimization.
>
> * **Why is random better than the Regularized Evolution:** As also discussed with Reviewer shXW, this can occur in combinatorial optimization when the search landscape has many poor local optima. An evolutionary algorithm can get trapped in one of these optima early on. A random search, by contrast, does not have this issue as it continues to explore the entire space throughout the search, which can be a surprisingly effective strategy in such cases.
>
> [1] Bayesian Optimization over Permutation Spaces (AAAI, 2022)
>
> [2] Combining Latent Space and Structured Kernels for Bayesian Optimization over Combinatorial Spaces (NeurIPS, 2021)
>
> [3] Understanding LLM Embeddings for Regression (TMLR, 2025)
>
> [4] Exploring the Limits of Transfer Learning with a Unified Text-to-Text Transformer (2019)
>
> [5] https://huggingface.co/datasets/allenai/c4

---

> ### Author Response · Authors · 2025-09-02
> **Response to Reviewer VPct, Part 2**
>
> * **Objective functions on Figure 5:** To test our method on more realistic blackbox optimization problems, we applied our method on actual production problems found in industry. However, finding the actual objective functions for online evaluation is extremely challenging (we would need the exact code, execution environment, etc. which also would’ve changed over time).
>     * Luckily, we have lots of offline evaluations $(x_1, f(x_1)), (x_2, f(x_2)), ...$ of these objectives. Thus we can instead use surrogate-based benchmarking, i.e. train a separate regressor (e.g. XGBoost) over offline evaluations, and consider it as a surrogate of the actual objective. Prior research [1, 2] has found this to be a sufficient yet cheap way of comparing algorithms.
>
> * **Other minor requests (citing, grammar)**: Fixed.
>
> [1] Surrogate NAS benchmarks: Going beyond the limited search spaces of tabular NAS benchmark
>
> [2] Efficient benchmarking of hyperparameter optimizers via surrogates

---

> > ### Comment · Reviewer_VPct · 2025-09-23
> > **Have you actually added problems typical for industry?**
> >
> > I am a little bit confused because you mention you have done that.
> >
> > Please, clarify, I think it's important for this work, given that it claims to be an empirical work, e.g. that it doesn't delve into an analysis of the embeddings (Ref: your comment above).

---

> ### Author Response · Authors · 2025-09-16
> **Gentle ping - 2 weeks later**
>
> Hi Reviewer VPct, following up since our last response 2 weeks ago - please let us know if we've resolved your concerns, and feel free to respond with further questions if needed.

---

> ### Author Response · Authors · 2025-09-23
> **Yes**
>
> Hi Reviewer VPct,
>
> To give more context around the production problems - behind the scenes, common to many industries, we run a large hyperparameter tuning service for multiple users - it can be setup with any open source package (e.g. Ax, OSS Vizier, Optuna, Wandb).
>
> Due to the current anonymity required during the TMLR review phase, we can't disclose all identifying information yet, on what exact service and users we have (but we'd be happy to, for the camera ready revision).
>
> In any case, we've collected historical tuning runs for many production-level optimization problems for surrogate benchmarking. While Figure 5 (top row) contains standard ML problems like ResNet50 on ImageNet, the bottom row contains private production metrics. We can't legally disclose full specifications, but to give a rough sense:
>  * **Phone Hardware (4P):** Tune the camera quality on a new phone design, where hyperparameters are specs including denoising filters, texture retention, and image sharpening.
> * **Production Metric A (5P):** Maximize hardware performance, by tuning architecture components like CPU clocking and memory cache settings.
> * **Production Metric B (6P):** Optimize the click-rate of a webpage by tuning its features (e.g. fontsizes, coloring, images).
> * **Production Metric C (3P):** Reduce latency of a data-center by tuning its bin-packing algorithm used for job scheduling.
>
> We hope this helps and demonstrates substantial empirical analysis on high-impact, real-world objectives.
>
> As for embedding analysis, we emphasize that [x] performed their analyses also on T5 models and very similar synthetic objectives - we'll end up with the exact same results and performing duplicate work.
>
> [x] Understanding LLM Embeddings for Regression (TMLR, 2025)

---

### Decision · Action_Editor_vayr · 2025-10-09

**Recommendation:** Reject

**Additional Comments:**

The paper is highly interesting and the proposed method is likely very good, but unfortunately the paper falls short of what is expected from a TMLR submission. The paper’s main strength is an innovative idea that has potential for follow-up work and is likely to be interesting, and from this perspective alone it would be a great paper, suitable for top-tier venues. By the TMLR evaluation standards, however, it is not quite there because the main shortcomings relate specifically to the TMLR evaluation criteria. The paper makes strong claims, stating a major one already in the title that was revised during the review process, but falls short in providing empirical evidence in such a way that the readers can understand when and why the method works. The experimentation is a series of quick demonstrations, each involving a range of engineering choices whose effects are not communicated for the readers, and there is no discussion on when and how the method might fail. The paper gives an impression the method is extremely good, sometimes outperforming the strongest baseline included by a massive margin, but we do not learn when this kind of performance can be expected.

The paper has also other shortcomings as a scientific paper, even though they were not brought up during the review. The introduction has no references except for one towards the end, there is no discussion on potential limitations of failure modes, and the methodological choices that likely influence the results strongly are provided without justification. To some extent these issues could be fixed in a minor revision, but they are still quite substantial modifications.

In summary, I do not think the paper is publishable in its current form and the modifications are too major to be carried out in a minor revision. However, I consider the paper in general of high value and would really like to see it published after making at least one of the experimental validation aspects more polished. One clear piece of solid empirical evidence that is reported according to the standards in the field would be enough, and the rest could still be there in the paper as long as it was very clearly indicated to be high-level preliminary demonstrations.

**Audience:**

Yes

**Audience Explanation:**

The paper proposes a clever method combining LLM-embeddings and in-context regression for solving Bayesian optimization problems and explains a relatively straightforward solution that is easy to understand. It is likely interesting for a broad audience by the virtue of presenting a new perspective to solving these problems and also because the method appears to work well. Also people applying these methods would be interested in the work, and some researchers could be interested in improving specific components of the overall approach.

However, as one of the reviewers elegantly formulated: “In my view, readers of TLMR would be interested in that study, but not in the findings. There is no understanding why the method works”. I agree with this; the authors fail to communicate the insights so that the paper would be interesting for the scientific community, as a work to build on.

After careful consideration I would still judge the paper to satisfy the evaluation criterion as it indeed communicates a scientific idea many in the community would find interesting, despite lacking in execution.

**Claims And Evidence:**

No

**Claims Explanation:**

The paper is borderline in this criterion, with disagreement between the reviewers. It includes broad empirical evidence that appears convincing, but it is not particularly accurate or clear due to lack of scientific rigour.

On a positive side, the paper includes broad range of empirical studies that are conducted on both established benchmarks and concrete use-cases, and ablation studies investigating the effect of some tuning choices are included. The experiments are very diverse and cover different kinds of scenarios, more than most Bayesian optimization papers do. Moreover, some of the empirical results are very strong, with the proposed method outperforming a relatively strong baseline. In other words, the paper presents empirical evidence that supports the claim that LLM-based representations and in-context regression models can be useful in Bayesian optimization.

However, the reporting of the empirical results is in general sloppy and not transparent, to the extent that the empirical evidence remains as high-level demonstration or perhaps even an advertisement. This is also highlighted by the claims themselves, which are not sharp and clearly specified but generic and vague (“can achieve … competitive with state-of-the-art methods”). Some issues, like use of only one ad-hoc choice of a baseline and lack of any experimentation that justifies specifically the use of LLMs as representations over lighter alternatives, were pointed out by the reviewers and were only partially addressed during the review process. Closer inspection reveals other shortcomings and lack of transparency, to the extent that the paper does not really meet the standards of reporting empirical machine learning work.

Perhaps the most obvious case is the experiment reported in Fig 3, but similar argumentation holds for the other experiments as well. The overall process where the model is trained on stratified collection of optimization problems with data augmentation is sound and innovative, but the experiment is a trivialized demonstration instead of a proper validation. All of the technical elements (use of data augmentation, specific forms of augmentation, number of training tasks, uniform sampling of values during training) are merely mentioned without any real justification, there are absolutely no ablation studies on these aspects (that are much more relevant for the work than the generic ablations over dimensionality of the embeddings and the model size in Section 4.2), the metric is not what one would expect for a Bayesian optimization paper, the confidence intervals have been compressed to just half of standard devision perhaps to exaggerate the differences, and all discussion of the results is omitted. For two of the tasks the proposed model appears massively better than the baselines (possibly because of poorly chosen y-axis), which is something that absolutely would warrant discussion -- could this be caused by a highly similar task in the training data, poor performance of the specific baseline used here, or something else? One reviewer asked about this but the authors only speculated on one possible reason, not relating it to factual observations or modifying the paper.

After a serious and careful consideration I must judge the paper to not meet the criterion. The empirical experimentation is arguably substandard for a scientific publication in general, and TMLR as a venue that explicitly emphasises robust validation of the claims has here a higher standard. I believe the paper does not match the spirit of TMLR, even though the breadth of the experimentation would in many other contexts compensate for the lack of rigour.

**Resubmission Of Major Revision:**

The authors may consider submitting a major revision at a later time.